# Prevalence Rate and Associated Risk Factors of Anaemia among under Five Years Children in Ethiopia

**DOI:** 10.3390/nu14132693

**Published:** 2022-06-28

**Authors:** Bereket Tessema Zewude, Legesse Kassa Debusho

**Affiliations:** 1Department of Statistics, College of Natural and Computational Science, Wolaita Sodo University, Wolaita Sodo P.O. Box 138, Ethiopia; 2Department of Statistics, College of Science, Engineering and Technology (CSET), University of South Africa (UNISA), Pretoria 0002, South Africa; debuslk@unisa.ac.za

**Keywords:** Anaemia, cumulative logit model, generalised linear model, Malaria, proportional odds model, pseudo-maximum-likelihood method, survey design features

## Abstract

**Background:** Anaemia is a condition characterised by a decrease in the concentration of haemoglobin (Hb) in the blood. Anaemia suffers under five years children about 47.4% and 67.6% worldwide and developing countries including Ethiopia, respectively. The aim of this study was to assess the prevalence rate and the associated socio-economic, geographic and demographic factors of anaemia status of under five years children in Ethiopia. **Methods:** The data for this study were obtained from the 2011 Ethiopia National Malaria Indicator Survey (EMIS 2011). A sample of 4356 under five years age children was obtained from three regional states of Ethiopia. Based on haemoglobin level, child anaemia status was ordered and takes an ordinal value as no anaemia, mild anaemia, moderate anaemia and severe anaemia, respectively. Ordinal logistic regression model, specifically the proportional odds model was used by considering with and without survey design features. **Results:** Of the 4356 complete cases, 2190 (50.28%) were male and 1966 (49.72%) were female children under five years old. The children overall mean (SD) age was 2.68 (1.21) years. It was observed that both the mean ages and their variabilities in the regions are approximately equal to the overall mean and variability. It was also observed that in Amhara, Oromiya and SNNP regions 72.28%, 67.99% and 73.63% of the children, respectively had no anaemia; 15.93%, 13.47% and 13.56% of the children, respectively had mild anaemia; 10.99%, 15.61% and 11.33% of the children, respectively had moderate anaemia; and only 0.81%, 2.93% and 1.49% had severe anaemia, respectively. The prevalence of severe child anaemia status was higher in Oromiya region compared to Amhara and SNNP regions, respectively. Our result indicates that age, use of mosquito net, malaria RDT outcome, type of toilet facility, household wealth index, region and median altitude were significantly related to child anaemia status. However, it was observed that some covariates were model dependent, for example household wealth index and type of toilet facility were not significant when considering survey features. **Conclusions:** Anaemia burden remains high particularly in developing countries. Controlling the burden of anaemia necessitates the formulation of integrated interventions which prioritise the highest risk groups including children under five years. The statistical model used in this paper identified individual, household and cluster level risk factors of child anaemia. The identified risk factors for example not having improved toilet facility in the dwelling where a child lived as well as poorest household wealth index suggest the policymakers should target to focus more on children from poor community. Further, the strong association between malaria infection and anaemia suggests that malaria preventative methods such as vector control methods namely, long-lasting insecticidal nets (LLINs) and indoor residual spraying of households with insecticides and including case diagnostic testing and treatment may be the most effective ways to reduce infections associated with anaemia. Such collective assessment approach may lead to more effective public health strategies and could have important policy implications for health promotion and for the reduction of health disparities.

## 1. Introduction 

Anaemia is a condition characterised by a decrease in the concentration of haemoglobin (Hb) in the blood, or alternately it is defined as a decrease (>2 SD) in haemoglobin compared with the mean value for age. According to the 2015 World Health Organisation (WHO), anaemia affects around 800 million children worldwide. Similarly, anaemia suffers an estimated 273.2 million or about 47.4% for under five years children worldwide and developing countries suffers from anaemia covers about 67.6% of children aged under five years [1,2,3]. It is also one of the major public health problems in Africa which is malaria endemic and one of the second world’s leading causes of disability [4,5]. A prevalence of 62.3% was from Sub-African countries affected from anaemia [6]. In Ethiopia, a lot of efforts are done to reduce the causes of anaemia. However, still it needs serious attention [7]. Therefore, in order to apply successful implementations to substantially reduce the burden of anaemia, there is a continuous need to understand the epidemiology and risk factors associated with the disease. There are a large number of studies worldwide that have identified a wide variety of risk factors; socioeconomic, geographic and demographic associated with anaemia (e.g., see [1,8,9,10,11,12,13,14,15,16,17,18,19,20,21,22,23,24,25,26,27]); however, there is still a great need to identify these factors in Ethiopia in order to implement successful strategy to control anaemia. Most recent studies mainly focus on primary data collected from hospital as well from household survey from a small group. However, our study mainly focuses on large representative national survey and this study is expected to provide great opportunity to the concerned bodies in formulating strategies and policies in general. The aim of this study is to assess the prevalence rate and the associated socio-economic, geographic and demographic factors of anaemia status of under five years of age children in Ethiopia. In this study, survey enumeration area (SEA) was considered as a cluster. The rest of the paper is organised as follows. The data and the statistical methods used for analyses are introduced in “Materials and Methods” section. The results from applying these methods to the study data are discussed in “Results” section. Finally, discussion, and conclusions and pointers for future study are given in “Discussion” and “Conclusions” sections and strengths and limitations of the study are given in “Strengths and Limitations” section, respectively.

## 2. Materials and Methods

### 2.1. Study Data 

The data used in this study were obtained from the 2011 Ethiopia National Malaria Indicator Survey (EMIS 2011). The survey was conducted by the Ethiopian Health and Nutrition Institutes and its partners, the Ethiopian Ministry of Health in collaboration with the Central Statistics Agency (CSA), US President’s Malaria Initiative (PMI), United Nations Children’s Fund (UNICEF), Malaria Control and Evaluation Partnership in Africa (MACEPA/PATH), Malaria Consortium, The Carter Centre (TCC), World Health Organisation (WHO), and International Centre for AIDS Care and Treatment Programs (ICAP). The EMIS was a large nationally representative survey designed to cover key malaria control interventions, treatment-seeking behaviour, malaria prevalence; and also to assess anaemia prevalence in children under five years of age, malaria knowledge among women and indicators of socioeconomic status, see [7]. The survey consisted of a two-stage sample design. The first stage involved selecting clusters from a list of SEAs covered in the 2007 Population Census, these areas made up the primary sampling units (PSUs). Data from three major regional states namely Amhara, Oromia and Southern Nations, Nationalities and Peoples’ (SNNP) were used in this study. A total of 335 SEAs where 93 from Amhara, 154 from Oromia and 88 from SNNP regions were included in the analyses. These three regions were considered due to the data sharing policy of Ethiopian Public Health Institute that since we are not part of EMIS project team we had limited access to the survey data for our research. However, these regions had covered 77.5% of the national enumeration areas analysed for the survey report.

### 2.2. Response and Explanatory Variables

The dependent or response variable in this study was childhood anaemia status which was grouped into four categories using the variable that measures haemoglobin (Hb) level in blood. During the survey, blood samples were taken from all children under five years of age in every sampled household and from persons of all ages in every fourth household per WHO guidelines. The blood sample of a child was taken after obtaining consent from residents and assistance of the parent/guardian of the child. Haemoglobin testing for anaemia was done using Hemocue Hb 201 analysers for children under five years. The Ethiopian malaria indicator survey (EMIS 2011) also examined the haemoglobin levels of children under five years. The mean haemoglobin value across Survey Enumeration Areas (SEAs) was 11.1 g/dL with a standard deviation of 0.1. Considering the wide range in altitude across surveyed areas and normal increases in haemoglobin levels among populations living at high altitude, anaemia was defined according to WHO classification. Based on Ethiopian malaria indicator survey [7] and definition to WHO specification of anaemia for children under five years [5,28], we have classified childhood anaemia status as no anaemia (>11.0 g/dL), mild anaemia (8.0–11.0 g/dL), moderate anaemia (5.0–8.0 g/dL) and severe anaemia (<5.0 g/dL), respectively. Therefore, the anaemia status scale has a multinomial data structure, in particular taking an order value.

### 2.3. Independent Variables

The independent variables or covariates considered in this study were selected based on previous studies which have been conducted at the global level. Independent variables used to explain anaemia status of under five years children. These potential determinant factors consisted of demographic, socioeconomic and geographical variables, which are expected to be correlated with anaemia status of children under five years of age. The demographic variables include age of a child and gender of a child, and family size of the household, the first two were collected at the individual level. Socio-economic variables were child malaria RDT test result (positive or negative) which was collected at the individual level, number of household members or household size, whether household had mosquito nets that can be used while sleeping or not which was a dichotomous variable with the categories yes or no; main source of drinking water which was a three-category variable with the categories unprotected, protected source and piped water; household toilet facility which was a three-category variable with the categories no facility, pit latrine and flush toilet; and household wealth index which was a five-category variable with the categories poorest, second, middle, fourth and richest. A SEA or cluster characteristics are a region which was a three-category variable with the categories Amhara, Oromiya and SNNP regional states; and median altitude in meters.

### 2.4. Statistical Model

When considering without survey design, features can assume a simple random sample of under five years of aged children from an infinite population. It can be suited to apply for example ordinal logistic regression that is cumulative logit model. However, when data came from complex survey including primary sampling units, strata and survey weights such as EMIS 2011, considering survey design feature have a great importance in identifying associated risk factors of child anaemia. Hence, in our study we consider both with and without survey features under the class of generalised linear model that is ordinal logistic regression model, specifically the proportional odds model and presented below. 

### 2.5. Cumulative Logit Model

The cumulative logit model is the most popular logistic regression model for ordinal responses. As the child anaemia status categories are ordered, we have considered the cumulative logit models with proportional odds assumption. This model also called the proportional odds (PO) model. The PO model is a generalisation of a binary logistic regression model when the response variable has more than two ordinal categories. Let us consider independent data pairs ($x_{i},Y_{i}$), i=1,…,n.  where $x_{i}={\left(x_{i1},\ldots ,x_{ip}\right)}^{\prime }$ are independent variables and $Y_{i}$ is ordinal response in our case child anaemia status. The cumulative logit is formed by using cumulative probabilities given by:(1)$$\pi_{ij}\left(x_{i}\right)=P\left(Y_{i}\leq j|x_{i}\right)=\overset{j}{\underset{h=1}{\sum }}\pi_{ij\;,\;}j=1,\ldots ,J$$

The cumulative logit models for ordinal responses can examine the effects of independent variables $x_{i}$ on the log-odds of cumulative probabilities and defined as follows ([4,29]).
(2)$$logit(P\left(Y_{i}\leq j|x_{i}\right)=log\left(\frac{\pi_{ij}\left(x_{i}\right)}{1-\pi_{ij}\left(x_{i}\right)}\right)=log\left(\frac{\sum_{h=1}^{j}\pi_{ij}}{\sum_{j=j+1}^{J}\pi_{ij}}\right)$$

Observe that each cumulative logit uses all *J* response categories. A model for $logit(P\left(Y_{i}\leq j|x_{i}\right)$ is an ordinary logistic regression model for a dichotomous response in which categories from 1 to *j* comprise the first outcome, whereas categories from *j* + 1 to *J* comprise the second one. The parsimonious model that employs all *J* − 1 cumulative logits at the same time has the following cumulative logit form [4,29].
(3)$$logit(P\left(Y_{i}\leq j|x_{i}\right)=log\left(\frac{\pi_{ij}\left(x_{i}\right)}{1-\pi_{ij}\left(x_{i}\right)}\right)=\alpha_{j}+x_{i}^{\prime }\beta ,\;\;j=1,\ldots ,J-1$$
where, $\alpha_{j}$ is the jth intercept coefficient for cummulative logit and $\beta ={\left(\beta_{1},\ldots ,\beta_{p}\right)}^{\prime }$ is regression parameters associated with vector of independent variables associated with $x_{i}$. This is the form of a proportional odds model because an odds ratio of any predictor variable is assumed to be constant across all categories. Here, the effects estimates (β) are negative since the cumulative logit model coefficients have the form *−β*. The $\alpha_{j}’$s are referred to as cut-points or thresholds. For fixed $x_{i}$, since $\pi_{ij}\left(x_{i}\right)$ increases in *j*, the s also increases in *j* and hence the logit in (3) is an increasing function of $\pi_{ij}\left(x_{i}\right)$. The regression parameters can be estimated using maximum likelihood method. The model in (3) is written with a negative sign in front of ***β*** so that a positive value for ***β*** then implies a positive relationship. Let us extend a cumulative logit model for an ordinal response in Equations (2) and (3) by considering the sample survey design features. Assume that y with K category is defined for the probability of having less than or equal to k, relative to the probability of having y greater than k, k=1,…,K. Using this definition for survey data, the cumulative logit regression model is given as:(4)$$logit\left(P(y\leq k|x)\right)=log\left(\frac{\sum_{u=1}^{k}P(y\leq k|x)}{\sum_{u=k+1}^{K}P(y>k|x)}\right)=\beta_{0\left(k\right)}-\left(\beta_{1},\ldots ,\beta_{p}\right)$$
where, x is a p×1 vector of independent variables, $\beta_{0\left(k\right)},\;k=1,\ldots ,K\;and\;\beta_{1},\ldots ,\beta_{p}$ are regression parameters estimated by maximizing pseudo maximum likelihood function. As in model (4), for each category k, the cumulative logit function includes a unique intercept $\beta_{0\left(k\right)}$, but all k categories share a common set of regression parameters for p independent variables, $\beta =(\beta_{1},\ldots ,\beta_{p}).$ Using the model (4), the kth cumulative probability can be given as:(5)$$\pi_{k}\left(x\right)=P\left(y\leq k|x\right)=\frac{\exp\left(x^{\prime }\beta \right)}{1+\exp\left(x^{\prime }\beta \right)},\;k=1,\ldots ,K-1$$

## 3. Results 

### 3.1. Descriptive Summary 

Table 1 presents about the prevalence of children anaemia status with different regions and distribution of independent variables for both continuous and categorical variables. For continuous independent variables, mean and standard deviation (SD) in brackets were used. Further, for categorical independent variables, number and percent were used.

Considering only the complete cases, a total of 4356 children were in this study, of which 2190 (50.28%) were male and 1966 (49.72%) were female. The distribution of children by region was 49.90, 50.50 and 50.14 percent male in Amhara, Oromia and SNNP, respectively (see Table 1). The children overall mean (SD) age was 2.68 (1.21) years and by region they were 2.62 (1.22), 2.67 (1.22) and 2.74 (1.18) years in Amhara, Oromia and SNNP, respectively. Observe that both the mean ages and their variabilities in the regions are approximately equal to the overall mean and variability. The average number of household residents, i.e., household size, was about six across the study areas and variability in household sizes was about 2 standard deviations. In Amhara, Oromiya and SNNP regions 50.91%, 62.88% and 54.41% of under five years of age children, respectively lived in the sampled households which had used unprotected drinking water supplies. These include unprotected dug well, unprotected spring, rainwater and irrigation channel. On the other hand, about 24.80%, 15.13% and 19.87% of under five years of age children in this study lived in the households which had protected drinking water supplies (which were protected dug well, protected spring, tanker truck and cart with small tank) in Amhara, Oromiya and SNNP, respectively. Only 24.29%, 21.99% and 25.72% of the children in Amhara, Oromiya and SNNP, respectively lived in the sampled households that used piped water as main source of drinking. The pipped water in the regions includes piped into dwelling, piped into yard or plot, public tap or standpipe and tube well or borehole. About 25.50% in Amhara, 8.57% in Oromiya and 34.82% in SNNP regions under five years of age children in this study came from households which had flush or hanging toilet facility. Again, SNNP region had the largest percent of children (53.11%) that came from sampled households which had either pit latrine toilet with slab or without slab or open pit toilet facilities compared to Amhara (36.29%) and Oromiya (46.13%). Results in Table 1 also show that the Oromiya region had the largest percent of under five years of age children (45.30%) that came from sampled households with other type of toilet facilities such as composting toilet, bucket toilet or no facility or bush or field; whereas 38.21% in Amhara and 12.07% in SNNP regions under five years of age children in the sampled households had such toilet facilities. The results in Table 1 also show that 75.71% in Amhara, 36.07% in Oromiya and 41.78% in SNNP regions children lived in the households that used mosquito nets. The largest percent of children in Amhara region (23.29%) were from households where their wealth status was under the fourth category, 22.88%, 19.66%, 17.44% and 16.73 percent of children in the same region were from households that had wealth status categories second, richest, middle and poorest, respectively. However, in Oromiya region, households of 29.51%, 21.47%, 16.97%, 16.44% and 15.61 percent children had the wealth status categories poorest, second, richest, fourth and middle, respectively. Whereas in the SNNP region, households of 25.44%, 23.58%, 20.43%, 18.20% and 12.35% children had the wealth status categories middle, second, richest, fourth and poorest, respectively. Table 1 also displays the prevalence of childhood malaria and anaemia status by regions. The results in this study show that 2.72%, 1.05% and 4.83% of under five years old children tested positive for malaria in Amhara, Oromiya and SNNP regions, respectively. It was also observed that in Amhara, Oromiya and SNNP regions 72.28%, 67.99% and 73.63% of the children, respectively had no anaemia; 15.93%, 13.47% and 13.56% of the children, respectively had mild anaemia; 10.99%, 15.61% and 11.33% of the children, respectively had moderate anaemia; and only 0.81%, 2.93% and 1.49% had severe anaemia, respectively. The severe child anaemia status was higher in Oromiya region compared to Amhara and SNNP regions, respectively. The mean (SD) median-altitude of these households in Amhara, Oromia and SNNP regions was 2033.120 m (408.830 m), 1904.560 m (377.618 m) and 1931.270 m (366.876 m), respectively.

### 3.2. Cumulative Logit Model for Anaemia Status of under Five Years Age Children 

As the child anaemia status is ordered, we considered the ordinal logistic regression model, specifically the proportional odds model was applied. The steps in model building for ordinal logistic regression models are as follows by considering with and without survey features. First, we have employed univariate as well as multivariable techniques to identify the factors that are associated with anaemia prevalence in under five years of age children. In the univariate analysis, a simple cumulative logit model was fitted between the potential risk factors and the anaemia status or category of a child. A Wald test was applied to test the significance of each of the potential risk factor in a univariate analysis. The significant risk factors were used to define the initial main-effects cumulative logit model, epidemiologically relevant interaction effects, e.g., an interaction between age and gender introduction before deciding on the final multivariable cumulative logit model. Again, the initial main effects multivariable cumulative logit model was fitted to the anaemia data and age, malaria RDT result, household size, median altitude, type of toilet facility, household wealth status and region were found to be significant predictors at 5% level. However, main source of drinking water was non-significant. The log-likelihood ratio chi-square test statistic *LRT*
$\chi_{14}^{2}=373.665$ with *p*-value < 0.0001, which indicates the null hypothesis that the covariates do not contribute to the model was rejected. Therefore, the fitted initial main effects cumulative logit model was better than the null model with no covariates in predicting the cumulative odds of being at or above a particular anaemia status category. We have also checked the delta beta hat percentages to ensure whether main source of drinking water has confounding effect on the final main effects multivariate proportional odds cumulative logit model. When we removed main source of drinking water from the initial main effects model, the largest percent changes that we found was 8.93% for the coefficient of pit latrine toilet category. As this delta beta hat percentage does not exceed our criterion of 20%, the main source of drinking water was not a confounder. Therefore, adding main source of drinking water to the initial multivariable proportional odds cumulative logit model might not provide important statistical adjustment to the effects of the variables that remained in the model. Finally, the Brant test of proportional odds assumption can be examined using the brant() function from brant package in R environment if the model was fitted with the polr() function from the MASS package. The software provides the univariate Brant test result for each predictor variable and the omnibus test for the overall model. The omnibus Brant test of proportional odds assumption yields ($\chi^{2}\left(18\right)=64.24$) with *p*-value < 0.0001 and it shows that the proportional odds assumption for the fitted final main effects multivariate cumulative logit model was violated. Whereas except for malaria RDT result and Amhara region the Brant tests for other predictor variables show that the proportional odds assumption was upheld (see Table 2). When the omnibus Brant test of proportional odds assumption has a small *p*-value, it is helpful to check whether the violation of the assumption is practically important by comparing estimates obtained from separate binary logistic models fitted to the binary collapsed response. When assessing the proportional odds assumption by considering survey features based on the design-based Wald test, proportional odds or equal slopes assumption statistic was *F*(14,316) = 1.91 with *p*-value = 0.0252. It suggests that the data do not support the proportional odds assumption. The design-based Wald tests for each predictor variable show that the proportional odds assumption was violated for gender (with *p*-value = 0.0125), malaria RDT outcome (with *p*-value = 0.0004), Amhara region (with *p*-value = 0.0008), type of toilet facility (for Flush and hanging toilets *p*-value = 0.0188 and for Pit latrine toilets *p*-value = 0.0057) and for interaction between age and gender (with *p*-value = 0.0022). Based on Equation (3), the initial main effects of multivariate cumulative logit model of proportional odds take the following form
$$logit(P\left(Y_{i}\leq j|x_{i}\right)=\alpha_{j}-\left(\beta_{1}Age+\ldots +\beta_{13}SNNP\right)+\beta_{14}med\;Alititude,\;j=1,2,3$$
and we call this model M1. Considering M1 model, the formulation of the final main effects multivariate cumulative logit models whose results presented in Table 3 for the no anaemia, mild, moderate categories become:$$\begin{array}{ll} \mathrm{logit}\left(P\left(Y\leq 1\right)\right)= & -2.4136+0.3255\mathrm{Age}+0.2253\mathrm{Net}-1.4967\mathrm{RDT}\;\mathrm{result}+0.0292\left(\mathrm{HH}\;\mathrm{size}\right)+0.1613\mathrm{Pit}\;\mathrm{latrine}\\ & +0.1946\mathrm{Flush}\;\&\;Hanging\;toilet+0.0776Second+0.0891Middle+0.4051Fourth\\ & +0.2937Richest+0.1192Amhara+0.2335SNNPR+0.0009Median\;altitude \end{array}$$
$$\begin{array}{ll} \mathrm{logit}\left(P\left(Y\leq 2\right)\right)= & -1.5254+0.3255\mathrm{Age}+0.2253\mathrm{Net}-1.4967\mathrm{RDT}\;\mathrm{result}+0.0292\left(\mathrm{HH}\;\mathrm{size}\right)+0.1613\mathrm{Pit}\;\mathrm{latrine}\\ & +0.1946\mathrm{Flush}\;\&\;Hanging\;toilet+0.0776Second+0.0891Middle+0.4051Fourth\\ & +0.2937Richest+0.1192Amhara+0.2335SNNPR+0.0009Median\;altitude \end{array}$$
$$\begin{array}{ll} \mathrm{logit}\left(P\left(Y\leq 3\right)\right)= & 0.7218+0.3255\mathrm{Age}+0.2253\mathrm{Net}-1.4967\mathrm{RDT}\;\mathrm{result}+0.0292\left(\mathrm{HH}\;\mathrm{size}\right)+0.1613\mathrm{Pit}\;\mathrm{latrine}\\ & +0.1946\mathrm{Flush}\;\&\;Hanging\;toilet+0.0776Second+0.0891Middle+0.4051Fourth\\ & +0.2937Richest+0.1192Amhara+0.2335SNNPR+0.0009Median\;altitude \end{array}$$

The final results for the final main effects multivariate cumulative logit model for the relationship of anaemia prevalence to predictor variables is presented in Table 3. 

The result indicates that age, use of mosquito net, malaria RDT outcome, type of toilet facility, household wealth index, region and median altitude were significantly related to child anaemia status (see Table 3). However, it was observed that some covariates were model dependent, for example household wealth index and type of toilet facility were not significant when considering survey features. Further, the interaction effect of age and gender was not significantly associated with the child’s anaemia status (see Table 4). The interpretation for regression parameters (${\hat{\beta }}_{l}$) and odds ratio (OR) are given below, and similar interpretations have been drawn when considering survey features. From Table 3 we can understand that except for malaria RDT result, the effects estimate ${\hat{\beta }}_{l},\;l=1,\ldots ,12$ are negative. The estimates for age, household size and median altitude are−0.3255, −0.0292 and −0.0009, respectively and suggest that the cumulative probability starting at “No anaemia” status to “Severe anaemia” status decreased as the age, household size and median altitude increase, in each case holding the effects of other explanatory variables constant. The estimate for age −${\hat{\beta }}_{1}$ = 0.3255 suggests that for under five years of age children the log odds of being in no anaemia status category versus mild or moderate increased by 0.3255 for a one year increase in age, holding the other explanatory variables constant. Similarly, the estimate for household size *−*${\hat{\beta }}_{3}$= 0.0292 suggests that for under five years of age child the log odds of being in no anaemia status category versus mild or moderate increased by 0.0292 when the household size or members of the household where the child lived increased by a member, holding the other explanatory variables constant and the log odds of being in no anaemia status category versus mild or moderate increased by 0.09 for a 100 m increase in median altitude, holding the other explanatory variables constant. Similarly, the coefficient 1.4967 indicates that for those children who tested positive for malaria based on the RDT result, the log odds of being no anaemia status versus mild or moderate was 1.4967, lower than those who tested negative, holding the other explanatory variables constant. The log odds of being in no anaemia status category versus mild or moderate for under five years of age children from second, middle, fourth and richest household wealth status are 0.0776, 0.0891, 0.4051 and 0.2937 higher than those children from poorest household wealth status, respectively holding the other explanatory variables constant. Whereas for those under five years of age children who lived in Amhara and SNNPR regions, the log odds of being in no anaemia status category versus mild or moderate were 0.1192 and 0.2335 higher than those who lived in Oromiya region. The odds ratio (OR) for age is 0.7222, which indicates that for a one-year increase in a child age, the odds of being in a better anaemia status decrease by 27.78%. For mosquito net use, the OR = 0.7983, indicates that there was a significant difference between those children who lived in household that had mosquito nests that can be used while sleeping and those who lived in households that do not in anaemia status, holding the other explanatory variables constant. For the type of toilet facility, the odds ratios are 0.8231 and 0.8510 for households that had pit latrine and flush and hanging toilets, respectively. These indicate that the odds of being in bad anaemia status for children living in these households were 17.69% and 14.9% lower than those children living in households without toilet facility, respectively. For the household wealth status, the odds ratios are 0.9253, 0.9148, 0.6669 and 0.7455 for second, middle, fourth and richest households, respectively. These indicate that the odds of being in bad anaemia status for children from these households are 7.47%, 8.52%, 33.31%, and 25.45% lower than those children from the poorest household, respectively. The OR for median altitude is 0.9990, which indicates that there was a significant linear relationship between median altitude and the cumulative odds being in bad anaemia status, holding the other explanatory variables constant. 

When considering survey features, except the coefficients for main effect of malaria RDT result effect and interaction effect between age and gender, the other coefficients are all negative (see Table 4). The estimated cumulative odds ratio for age is $({\hat{OR}}_{y\leq k:age}=0.682$ (with 95% CI: (0.628, 0.741)), suggesting that the cumulative odds of being in a category with higher severity of anaemia relative to a category with less severity of anaemia were decreased by 31.8% for each additional year of age holding the other predictors fixed. The estimated cumulative odds ratio that a female child with male child was $({\hat{OR}}_{y\leq k:Female}=0.718$ (with 95% CI: (0.538, 0.959)). The cumulative odds that a female child was in a category with higher severity of anaemia relative to a category with less severity of anaemia are 71.8% of those same odds for a male child with the same age, malaria RDT result, household size, mosquito net use, toilet facility, household wealth status, region and median altitude. We note that female and male children differ significantly in terms of distributions on anaemia status when holding other predictors fixed. The odds of anaemia were significantly higher for children who tested positive for malaria based on RDT result compared to those who tested negative $({\hat{OR}}_{y\leq k:RDT\;positive}=4.515$; 95% CI: (2.779, 7.338)). The estimated cumulative odds ratios for mosquito net use, household size and median altitude are ${\hat{OR}}_{y\leq k:Mosquito\;Net\;Use}=0.765\;{\hat{OR}}_{y\leq k:Household\;size}=0.971\;{\hat{OR}}_{y\leq k:Median\;Altitude}=0.999$, respectively. These odds ratios suggest that the cumulative odds of being in a category with higher severity of anaemia relative to a category with less severity of anaemia are decreased by 2.9% for each additional household member, and decreased by 23.5% for children living in the households that had mosquito nets that can be used while sleeping compared to those who did not have and decreased by 1% for 1000 m increase in median altitude. The cumulative odds of being in a category with higher severity of anaemia relative to a category with less severity of anaemia are lower in Amhara $({\hat{OR}}_{y\leq k:Amhara}=0.856$ with 95% CI: (0.668, 1.096)) and SNNPR (${\hat{OR}}_{y\leq k:SNNPR}=0.750$ with 95% CI: (0.590, 0.953)) regions relative to the Oromiya region. Recall that the interaction effect of age and gender was not significantly associated with the child’s anaemia status. Children in households with improved toilet facilities had lower odds of anaemia compared to those in households with no toilet facilities, ${\hat{OR}}_{y\leq k:Pit\;latrine}=0.805$ with 95% CI: (0.644, 1.007) for PIT latrine and ${\hat{OR}}_{y\leq k:Flush\;toilet}=0.743$ with 95% CI: (0.563, 0.980) for flush toilet. The results in Table 4 also show that the odds of being in a category with higher severity of anaemia relative to a category with less severity of anaemia for children from the four categories of household wealth status, i.e., second, middle, fourth and richest are decreased by about 7.6%, 9.8%, 27.6% and 25.5%, respectively relative to children from poorest household.

## 4. Discussion 

The response variable is anaemia status of under five years of age child, which was grouped into four categories using the variable that measures haemoglobin (Hb) level in blood. The four categories for the child anaemia status are namely no anaemia (>11.0 g/dL), mild anaemia (8.0–11.0 g/dL), moderate anaemia (5.0–8.0 g/dL) and severe anaemia (<5.0 g/dL) as per the specification of WHO and EMIS 2011. Hence, the anaemia status scale has an ordinal data with an order. Different literatures reviewed the independent variables including age of a child, gender of child, child malaria RDT result (tested positive or negative), number of household members, household had mosquito nets, main source of drinking water, household toilet facility, household wealth index, region and median altitude in meters. In this study, we attempted to assess if these predictors or covariates significantly predict the anaemia status of under five years of age child. Our result indicates that age, use of mosquito net, malaria RDT outcome, type of toilet facility, household wealth index, region and median altitude were significantly related to child anaemia status (see Table 3). However, it was observed that some covariates were model dependent, for example household wealth index and type of toilet facility were not significant when considering survey features. Further, the interaction effect of age and gender was not significantly associated with the child’s anaemia status (see Table 4). In this study, the demographic factors age is one of the important factors for determining child anaemia status. The result shows that for a one year increase in a child age, the odds of being in a better anaemia status decrease. This finding agrees with the previous studies conducted for under five years children for example [1,9,11,12,15,16,22,23,26] they indicated that there was an association between age and anaemia status. This might be due to the haemoglobin concentration had undeviating and a progressive association with age. It is believed that as age increased, an insistent dietary nutrient for growth relatively becomes lower than an early age. This might be also attributable to lack of fulfilment for increased requirement of necessary nutrients for haemoglobin synthesis in age groups.

Moreover, demographic factors socio-economic factors play a vital role in determining child anaemia status. The finding indicates that there was a significant difference between those children who lived in household that had mosquito nets that can be used while sleeping and those who lived in households that do not have mosquito nets and this finding consistent with [24]. Similarly, the odds of being in bad anaemia status for children living in household’s toilet with pit latrine and flush and hanging toilet were lower than those children living in households without toilet facility. This finding is consistent with studies conducted in different countries, for example [10,11,12,17], that suggested that type of toilet facility was found to be significantly associated with anaemia status. The reason might be poor local sanitation *causes* lower haemoglobin levels and higher rates of anaemia in children. Based on Table 3 and Table 4, this study shows that the odds of anaemia were significantly higher for children who tested positive for malaria based on RDT outcome compared to those children who tested negative. This finding agreed with previous studies [10,24,27] that found that malaria RDT positive and anaemia status are significantly associated. This might be that being malaria positive reduces immunity and nutrient content in the body, leading to anaemic condition. Further, the result shows that household wealth index is significantly associated with child anaemia status. The poorest household status is more vulnerable to anaemic condition compared to middle, fourth and richest wealth status. This finding is consistent with the finding obtained from [9,10,11,14,16,17,20,23,24,30,31]. This is due to the reason that children from poor households are less likely to get iron-rich foods such as animal foods and vitamin-rich foods especially vitamins A and C which are very important for iron absorption.

Geographic factors are also related to child anaemia status, for example the odds of being in anaemia status category that show anaemia infection of a child were lower in Amhara and SNNPR regions compared to children living in Oromiya region. This finding agreed with [10] and disagreed with [14] which was carried out in Ethiopia. Moreover, the result indicates that there was a significant linear relationship between median altitude and the cumulative odds of being in bad anaemia status. This finding is consistent with pervious study conducted in Peruvian [8]. The reason could be children at high altitudes had significantly lower oxygen saturation on presentation and higher haemoglobin.

### Strengths and Limitations of the Study

This is the first article since most recent studies mainly focused on primary data collected from hospital as well as from household survey from a small group. However, our study analyses the complex survey design which is collected by the Ethiopian Public Health Institute (EPHI) and its partners which consider with and without sample survey features with large and national representative survey data set that comprehensively estimates the burden of anaemia and its associated factors among under five years children in Ethiopia, which could be taken as the strength of the study. However, this study has limitation due to absence of some information in the study data. In the 2011 Ethiopia malaria indicator survey, anaemia testing was limited to haemoglobin; no further information on types of anaemia is available in the study data. Since nutritional information or iron status of children and information on children’s infectious status or morbidity were not available, we could not assess in our study the effect of iron deficiency and infectious morbidity except malaria on childhood anaemia. Therefore, it is sensible to investigate further to substantiate the results obtained in the current study.

## 5. Conclusions

Globally, anaemia is a major public health issue, but varies considerably by country such as Ethiopia. Anaemia burden remains high particularly in developing countries. Controlling the burden of anaemia necessitates the formulation of integrated interventions which prioritise the highest risk groups including children under five years. The statistical model used in this paper identified individual, household and cluster level risk factors of child anaemia. The identified risk factors for example not having improved toilet facility in the dwelling where a child lived as well as poorest household wealth index suggest the policymakers should target to focus more on children from poor community. Further, the strong association between malaria infection and anaemia suggests that malaria preventative methods such as vector control methods namely, long-lasting insecticidal nets (LLINs) and indoor residual spraying of households with insecticides and including case diagnostic testing and treatment may be the most effective ways to reduce infections associated with anaemia. Such collective assessment approach may lead to more effective public health strategies and could have important policy implications for health promotion and for the reduction of health disparities. In future perspectives, including nutritional information or iron status of children and information on children’s infectious status or morbidity have to be considered, since the most common cause of anaemia globally is iron deficiency where approximately half of anaemia cases worldwide are attributable to iron deficiency and also, non-iron deficiency causes for example due to parasitic diseases, chronic infections such as cancer, tuberculosis and human immunodeficiency virus (HIV), hookworm infections, schistosomiasis and inherited or acquired disorders that affect haemoglobin synthesis, red blood cell production or red blood cell survival.

## Figures and Tables

**Table 1 nutrients-14-02693-t001:** Descriptive statistics for study variables.

			Region		
Variables		Amhara	Oromiya	SNNP	Total
Age (years)	mean (SD)	2.617 (1.216)	2.671 (1.222)	2.742 (1.181)	2.677 (1.211)
Household size	mean (SD)	5.526 (1.902)	5.832 (1.987)	5.905 (2.090)	5.781 (1.999)
Median alt.	mean (SD)	2033.12 (408.83)	1904.56 (377.62)	1931.27 (366.9)	1940.44 (385.74)
Child malaria RDT	Positive, N (%)	27 (2.72)	24 (1.05)	52 (4.83)	103 (2.36)
	Negative, N (%)	965 (97.28)	2263 (98.95)	1025 (95.17)	4253 (97.64)
Gender	Male, N (%)	495 (49.50)	1155 (50.50)	540 (50.14)	2190 (50.28)
	Female, N (%)	497 (50.10)	1132 (49.50)	337 (49.86)	1966 (49.72)
Main source of drinking water	Unprotected, N (%)	505 (50.91)	1438 (62.88)	586 (54.41)	2529 (58.06)
	Protected, N (%)	246 (24.80)	346 (15.13)	214 (19.87)	806 (18.50)
	Piped water, N (%)	241 (24.29)	503 (21.99)	277 (25.72)	1021 (23.44)
Type of toilet facility	Others, N (%)	379 (38.21)	1036 (45.30)	130 (12.07)	1545 (35.47)
	Pit latrine, N (%)	360 (36.29)	1055 (46.13)	572 (53.11)	1987 (45.62)
	Flush and hanging toilets, N (%)	253 (25.50)	196 (8.57)	375 (34.82)	824 (18.92)
Household used mosquito nets	No, N (%)	241 (24.29)	1462 (63.93)	627 (58.22)	2330 (53.49)
	Yes, N (%)	751 (75.71)	825 (36.07)	450 (41.78)	2026 (46.51)
Household wealth status	Poorest, N (%)	166 (16.73)	675 (29.51)	133 (12.35)	974 (22.36)
	Second, N (%)	227 (22.88)	491 (22.47)	254 (23.58)	972 (22.31)
	Middle, N (%)	173 (17.44)	357 (15.61)	274 (25.44)	804 (18.46)
	Fourth, N (%)	231 (23.29)	376 (16.44)	196 (18.20)	803 (18.43)
	Richest, N (%)	195 (19.66)	388 (16.97)	220 (20.43)	803 (18.43)
Child anaemia status	No anaemia, N (%)	717 (72.28)	1555 (67.99)	793 (73.63)	3065 (70.36)
	Mild, N (%)	158 (15.93)	308 (13.47)	146 (13.56)	612 (14.05)
	Moderate, N (%)	109 (10.99)	357 (15.61)	122 (11.33)	588 (13.50)
	Severe, N (%)	8 (0.81)	67 (2.93)	16 (1.49)	91 (2.09)

**Table 2 nutrients-14-02693-t002:** The Brant test of parallel regression assumption results for fitted initial main effects cumulative logit model for anaemia data.

Predictor	Category	Df	$\chi^{2}$	*p*-Value
Age (years)		2	1.69	0.4300
Malaria RDT outcome	Positive	2	10.24	0.0060
Mosquito net use	Yes	2	1.60	0.4490
Household size		2	5.77	0.0560
Type of toilet facility	Pit latrine	2	1.33	0.5160
	Flush & hanging toilets	2	4.31	0.1160
Household wealth status	Second	2	1.23	0.5400
	Middle	2	1.89	0.3890
	Fourth	2	1.50	0.4720
	Richest	2	2.58	0.2750
Region	Amhara	2	21.84	<0.0001
	SNNP	2	3.68	0.1580
Median altitude		2	3.90	0.1430
All predictors model		26	64.24	<0.0001

**Table 3 nutrients-14-02693-t003:** Results of final multivariate cumulative logit model for the relationship of anaemia prevalence to predictor variables without considering survey features.

Predictor	Category	$\hat{\beta }$	$\mathrm{se}(\hat{\beta })$	t-Value	OR	95% CI for OR	*p*-Value
Intercept	1–2	−2.4136	1.3922	−173.367			<0.0001
	2–3	−1.5254	0.0365	−41.759			<0.0001
	3–4	0.7218	1.0605	6.8060			<0.0001
Age (years)		−0.3255	0.0271	−12.0330	0.7222	(0.6848,0.7615)	<0.0001
Mosquito net use	Yes	−0.2253	0.0706	−3.1920	0.7983	(0.6952,0.9168)	0.0014
Malaria RDT outcome	Positive	1.4967	0.0069	217.4010	4.4668	(4.4069,4.5274)	<0.0001
Household size		−0.0292	0.0157	−1.8660	0.9712	(0.9418,1.0015)	0.0620
Type of toilet facility	Pit latrine Flush& hanging toilet	−0.1613 −0.1946	0.0692 0.0923	−2.3300 −2.1090	0.8231 0.8510	(0.6869,0.9864) (0.7430,0.9748)	0.0198 0.0349
Household wealth status	Second Middle	−0.0776 −0.0891	0.0684 0.0733	−1.1350 −1.2150	0.9253 0.9148	(0.8091,1.0581) (0.7923,1.0562)	0.2565 0.2243
	Fourth Richest	−0.4051 −0.2937	0.0761 0.0701	−5.3260 −4.1920	0.6669 0.7455	(0.5745,0.7741) (0.6498,0.8553)	<0.0001 <0.0001
Region	Amhara SNNPR	−0.1192 −0.2335	0.0929 0.0880	−1.2820 −2.6520	0.8876 0.7918	(0.7397,1.0651) (0.6663,0.9409)	0.1997 0.0079
Median altitude		−0.0009	0.0001	−15.8580	0.9990	(0.9989,0.9992)	<0.0001

**Table 4 nutrients-14-02693-t004:** Results for the final cumulative logit model for the relationship of anaemia prevalence to predictor variables considering survey features.

Predictor	Category	$\hat{\beta }$	$\mathrm{se}(\hat{\beta })$	t-Value	$OR_{y\leq k:j}$	$95\%\;\mathrm{CI}\;OR_{y\leq k:j}$	*p*-Value
Intercept	1–2	−2.8125	0.3391	−8.2950			<0.0001
	2–3	−1.9268	0.3559	−5.4140			<0.0001
	3–4	0.3469	0.4024	0.8620			0.3887
Age (years)		−0.3829	0.0420	−9.1160	0.6820	(0.6280,0.7410)	<0.0001
Gender Age × Gender Mosquito net use	Female Yes	−0.3308 0.0963 −0.2684	0.1469 0.0512 0.1047	−2.2510 1.8820 −2.5640	0.7180 1.1010 0.7650	(0.5380,0.9590) (0.9960,1.2180) (0.6220,0.9390)	0.0244 0.0598 0.0103
Malaria RDT outcome	Positive	1.5075	0.2468	6.1090	4.5150	(2.7790,7.3380)	<0.0001
Household size		−0.0293	0.0211	−1.3890	0.9710	(0.9320,1.0120)	0.1647
Type of toilet facility	Pit latrine Flush& hanging toilet	−0.2165 −0.2972	0.1134 0.1408	−1.9080 −2.1110	0.8050 0.7430	(0.6440,1.0070) (0.5630,0.9800)	0.0564 0.0347
Household wealth status	Second Middle	−0.0359 −0.0224	0.1295 0.1254	−0.2770 −0.1790	0.9650 0.9780	(0.7480,1.2450) (0.7640,1.2510)	0.7818 0.8582
	Fourth Richest	−0.3189 −0.1994	0.1499 0.1400	−2.1280 −1.4240	0.7270 0.8190	(0.5410,0.9760) (0.6220,1.0790)	0.0334 0.1544
Region	Amhara SNNPR	−0.1558 −0.2873	0.1258 0.1217	−1.2380 −2.3600	0.8560 0.7500	(0.6680,1.0960) (0.5900,0.9530)	0.2157 0.0183
Median altitude		−0.0010	0.0002	−5.7200	0.9980	(0.9986,0.9993)	<0.0001

## Data Availability

Not applicable.

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
