# Peer review of "Prevalence Rate and Associated Risk Factors of Anaemia among under Five Years Children in Ethiopia"

_nutrients, 2022, doi:10.3390/nu14132693_

Round 1

Reviewer 1 Report

I would like to congratulate the authors for writing this manuscript.

It is really rich and well designed.

I ask the authors to elaborate in their conclusion and include future perspectives 

Also, I ask the authors to include limits and strengths of the study

Author Response

Reviewer 1 Comments

I ask the authors to elaborate in their conclusion and include future perspectives. Also, I ask the authors to include limits and strengths of the study.

Authors’ Response:

We have considered the reviewer recommendation and revised conclusion and future perspectives. This is explained under abstract section on page 1 and also, under conclusion section see on page 17 and 18. We have also added strengths and limitations of the study section. See under strengths and limitations section on page 17 on revised manuscript with track changes.

Reviewer 2 Report

First of all English language was very poor, and  it was impossible to catch the actual meaning of some statements.

The scientific soundness of the paper relies on the statistical analysis of data, and I am not qualified at all to evaluate this aspect of the study.

Some suggestions to improve the paper:

  1. the authors graduated severity of anemia, based on Hb levels, but some values are missing: how do they categorize Hb levels from 8.0 to 9.9 g/dl?
  2. is the malaria status one of the main variables related to anemia status?  maybe other variables, such as nets in households and altitude are related to malaria; I could not catch this eventual correlation, maybe due to my incompetence in statistics, maybe due to poor english, maybe due to poor presentation of data
  3. why nutritional issues were not considered?  inadequate diet is one of the main causes of iron deficiency, which, in turn, is the major cause of anemia worldwide

Author Response

Reviewer 2 Comments

Some suggestions to improve the paper:

  1. The authors graduated severity of anemia, based on Hb levels, but some values are missing: how do they categorize Hb levels from 8.0 to 9.9 g/dl?

 Author’s response:

We have made corrections on reviewer comment. Response variable subsection under materials and methods section see page 5.

  1. Is the malaria status one of the main variables related to anemia status?  may be other variables, such as nets in households and altitude are related to malaria; I could not catch this eventual correlation, maybe due to my incompetence in statistics, maybe due to poor English, maybe due to poor presentation of data.

Author’s response:

We have made necessary corrections to address the comments. See under discussion section on page 16. Also, see under conclusion section on page 17 and 18.

  1. Why nutritional issues were not considered?  Inadequate diet is one of the main causes of iron deficiency, which, in turn, is the major cause of anemia worldwide.

Author’s response:

We have made necessary changes and added strengths and limitations section. See on page 17.

Reviewer 3 Report

Major points

- Introduction. I don't understand the definition “Anaemia is a condition characterized by a decrease in the concentration of hemoglobin (Hb) in the blood or alternately”: what’s the meaning for “alternately”? Maybe it would be more correct to define anemia as decrease (> 2 SD) in hemoglobin compared with the mean value for age; lines 48-49: “in Ethiopia, a lot of efforts done to reduce the anaemia disease”: I suggest that it would be better to write “the causes of anemia”.

- Materials and Methods. I don't understand why only the 3 largest regions of Ethiopia were analyzed. “Response variable”: the definition of anemia varies according to the age of the patient: a child who is less than 5 years old is anemic if the Hb is less than 11 g/dl, but under 6 months Hb is normal up to 9.5 g/dl; therefore, the population should probably be analyzed by age group (infant, < 6 months, < 2 years, < 5 years). “Statistical model”: I am not an expert in statistics, so I cannot comment on this, but probably the related part is too extensive and detailed and could be scaled down.

- Results. In table 1, I suggest adding “years” in age. In “gender”, the sum of the number of female is not 2166, but 1966, so these data gift to be revised. I think the numbers should be rounded both in the text and in the table, for example mean age of children. Table 1 does not include the prevalence of childhood malaria, as specified in the text (lines 200-201). Even in the results, in my opinion, the statistical part is too detailed and complicated, and for those who are not familiar with it, it is almost incomprehensible: I suggest summarizing and making the concept clearer for non-statisticians.

- Discussion. Perhaps, it should be better explained why age, altitude, and socioeconomic conditions influence a child's anemia status.

Author Response

Reviewer 3 Comments

Major points

- Introduction. I don't understand the definition “Anaemia is a condition characterized by a decrease in the concentration of hemoglobin (Hb) in the blood or alternately”: what’s the meaning for “alternately”? Maybe it would be more correct to define anemia as decrease (> 2 SD) in hemoglobin compared with the mean value for age; lines 48-49: “in Ethiopia, a lot of efforts done to reduce the anaemia disease”: I suggest that it would be better to write “the causes of anemia”.

 Author’s response:

We have made rephrased and corrected it. See under introduction section on page 3 in revised manuscript with track changes.

- Materials and Methods. I don't understand why only the 3 largest regions of Ethiopia were analyzed. “Response variable”: the definition of anemia varies according to the age of the patient: a child who is less than 5 years old is anemic if the Hb is less than 11 g/dl, but under 6 months Hb is normal up to 9.5 g/dl; therefore, the population should probably be analyzed by age group (infant, < 6 months, < 2 years, < 5 years). “Statistical model”: I am not an expert in statistics, so I cannot comment on this, but probably the related part is too extensive and detailed and could be scaled down.

Author’s response:

We have made corrections based on the comments.  Study data subsection under materials and methods section on page 4. Response variable subsection under materials and methods section on page 5. Statistical model subsection under materials and methods subsection  see on page 6.

- Results. In table 1, I suggest adding “years” in age. In “gender”, the sum of the number of female is not 2166, but 1966, so these data gift to be revised. I think the numbers should be rounded both in the text and in the table, for example mean age of children. Table 1 does not include the prevalence of childhood malaria, as specified in the text (lines 200-201). Even in the results, in my opinion, the statistical part is too detailed and complicated, and for those who are not familiar with it, it is almost incomprehensible: I suggest summarizing and making the concept clearer for non-statisticians.

Author’s response:

We have made corrections based on reviewer comments both in table and text adding “years “ and “gender” sum under abstract section on page 1, results section on page 8 (table 1) and 11(table 2 and 3) and 14(table 4). Similarly, we have made corrections for prevalence of childhood malaria on page 8 (table 1) under results section. We have also corrected making concept clearer for non-statisticians see on page 12 and 13.

- Discussion. Perhaps, it should be better explained why age, altitude, and socioeconomic conditions influence a child's anemia status.

Author’s response:

We have considered corrections based on reviewer comments. For age explanations see on page 1 under abstract section, on page 15 under discussion section. For altitude explanation see on page 17 under discussion section and for socioeconomic explanation see on page 16.

Round 2

Reviewer 2 Report

no specific comments

Author Response

We have submitted revised manuscript for MDPI language editing services and waiting their responses.

Reviewer 3 Report

Major points

- Introduction.alternately it is defined as a condition in which the number of red blood cell
is insufficient to meet physiologic needs”: This is not the correct definition of anemia. Anemia is defined as decrease (> 2 SD) in hemoglobin compared with the mean value for age.

- Materials and Methods.Statistical model”: I am not an expert in statistics, so I cannot comment on this, but probably the related part is too extensive and detailed and could be scaled down.

- Results. Even in the results, in my opinion, the statistical part is too detailed and complicated, and for those who are not familiar with it, it is almost incomprehensible: I suggest summarizing and making the concept clearer for non-statisticians.

Author Response

Major points

- Introduction. “alternately it is defined as a condition in which the number of red blood cell is insufficient to meet physiologic needs”: This is not the correct definition of anemia. Anemia is defined as decrease (> 2 SD) in hemoglobin compared with the mean value for age.

Author’s response:

We have made correction. See under introduction section on page 3 in revised manuscript with track changes.

- Materials and Methods.Statistical model”: I am not an expert in statistics, so I cannot comment on this, but probably the related part is too extensive and detailed and could be scaled down.

Author’s response:

We have made statistical models as much as possible to make clear and added under materials and methods section, subsection “cumulative logit model for anaemia status of under five years age children” (see page 5-7) and “considering survey features” (see page 15-16). Since we have considered different models, it is difficult to scale up and one step in the model is related to the next step.

Results. Even in the results, in my opinion, the statistical part is too detailed and complicated, and for those who are not familiar with it, it is almost incomprehensible: I suggest summarizing and making the concept clearer for non-statisticians.

Author’s response:

As much as possible, we have tried to make the concept clearer for non-statisticians. For example see page on (10-12) and (15-16).